# COMBINATION GENERALIZATION OF CAPABILITY-SPECIFIC NEURONS IN LLMS

## ABSTRACT

Although Large Language Models (LLMs) exhibit various exciting capabilities, understanding the mechanisms behind these abilities remains a challenging problem. In this paper, we aim to understand these mechanisms from the perspective of neurons. Specifically, we first propose a Detecting Capability-Specific Neurons (DCSN) method. Extensive enhancement and erasure experiments demonstrate that the detected neurons are highly correlated with specific capabilities, exhibiting strong cohesion and separability, which we define as capability-specific neurons. Moreover, leveraging these neurons, we conducted compositional experiments and, for the first time, discovered that capability neurons exhibit compositional generalization. Inspired by these findings, we propose a Capability Neuron-Level Fine-tuning method (CNLF) that fine-tunes specific capability neurons to achieve performance improvements across datasets and tasks. Extensive experiments validate the effectiveness of this method and provide a low-cost, highly generalizable fine-tuning paradigm. Our research offers interpretable insights into the capability mechanisms of LLMs.

## 1 INTRODUCTION

Large Language Models (LLMs) have demonstrated remarkable performance improvements across various natural language processing tasks (Zhao et al., 2023). Despite their powerful capabilities, the underlying principles of the mechanisms driving these capabilities (Yan et al., 2024; Haltaufderheide & Ranisch, 2024), as well as the relationship between model parameters and performance, remain unclear to humans (Peng et al., 2024). Recently, many studies have attempted to further understand and enhance the capabilities of these models, but their efforts have been hindered by the black-box nature of LLMs (Bonaldi et al., 2024; Sun et al., 2024). Therefore, understanding the internal mechanisms and characteristics of these models is key to improving their capabilities and interpretability (Ding et al., 2023).

Previous studies have attempted to establish a correspondence between knowledge or tasks and model parameters, defining these parameters as knowledge neurons or task neurons (Yao et al., 2024). However, the assumption of associating knowledge or tasks with specific neurons has been questioned in existing research (Dai et al., 2021). Studies have found a high degree of overlap among different knowledge neurons, which does not align with the expected localization of parameters (Huang et al., 2025b). Furthermore, the functions of these overlapping neurons remain unexplained (Huang et al., 2025a). Similar issues of inexplicability also exist with task neurons (Leng & Xiong, 2025).

Previous research has found that a model's capabilities can transcend knowledge and tasks, providing a potential explanation for overlapping neurons (Huang et al., 2025b). Achieving parameter localization of capabilities is key to interpretability research for LLMs (Trimmer, 2015). Inspired by these studies, we attempt to pose three questions: (1) Do neurons related to specific capabilities exist in LLMs? (2) Do capability neurons exhibit compositional generalization? (3) Can we improve LLMs through these neurons?

To address the three questions above, we conducted an analysis of capability neurons in LLMs. First, we constructed a compositional generalization dataset encompassing four computational capabilities: addition, subtraction, multiplication, and division. We then proposed a Detecting Capability-Specific Neurons (DCSN) method. Next, we performed enhancement and erasure experiments, demonstrating that these neurons are highly correlated with the respective capabilities. Our experiments show

that different capability neurons exhibit a low overlap rate (Separation), while neurons identified utilizing different datasets that demonstrate the same capability exhibit a high overlap rate (Cohesion) (MacQueen, 1967). Therefore, we refer to these as capability-specific neurons.

Utilizing the identified capability-specific neurons, we delved into the mechanism of capability invocation within the model. When performing tasks, the model often invokes multiple capabilities. To reflect this phenomenon, we constructed compositional problems within the compositional generalization dataset. For example, the problem "1 + 3 * 5 = ?" requires both addition and multiplication capabilities from the model. We collected neuron activation data during the model's execution of such tasks. Interestingly, the addition-specific neurons and multiplication-specific neurons were significantly activated. We then conducted extensive experiments across various problem types (involving different operations). The results demonstrated that capability-specific neurons exhibit compositional generalization, marking the first time the mechanism of capability invocation within the model has been revealed. Counterintuitively, we observed that when addition neurons were activated, subtraction neurons were activated 22% more than other neurons (e.g., for multiplication or division). This suggests a certain degree of association between addition and subtraction. Through ablation experiments, we inferred that this is likely due to the inverse relationship between addition and subtraction.

To enhance the model's various capabilities, we proposed a Capability Neuron-Level Fine-tuning method (CNLF). By fine-tuning these designated capability-specific neurons, we improved the model's capabilities and achieved superior performance across 12 downstream tasks. Compared to fine-tuning all parameters, this method improved performance by **18.9%** on unseen datasets. Additionally, we were able to control the model's individual capabilities, thereby enhancing the model's safety and controllability.

To the best of our knowledge, we are the first to discover that the model's capability-specific neurons possess compositional generalization. We also proposed a low-cost, highly generalizable fine-tuning method that enables control over the model's capabilities. This work sheds light on the internal mechanism of capability invocation within the model and enhances its interpretability. Our contributions can be summarized as follows:

- We proposed a Detecting Capability-Specific Neurons (DCSN) method, and successfully identified capability-specific neurons. Compared to previously studied knowledge neurons and task neurons, capability-specific neurons demonstrate superior separability and cohesion.
- To clarify the mechanism of capability invocation within the model, we conducted compositional generalization experiments and discovered that capability-specific neurons exhibit compositional generalization. This provides important insights for understanding and enhancing the model's multiple capabilities.
- We introduced a Capability Neuron-Level Fine-tuning method (CNLF), which simultaneously improves multiple capabilities of the model and achieves significant performance gains. Extensive experiments have demonstrated the effectiveness of the method.

## 2 RELATED WORK

**Parameter Localization.** Knowledge Neurons indicate that knowledge (such as triplets) can be localized in parameters, with storage forms including distributed parameters (Liu et al., 2024b), parameter layers (Meng et al., 2022b), and parameter chains (Yao et al., 2024). Task Neurons indicate that different tasks can be localized and utilize Capability Neurons for task generalization (Leng & Xiong, 2025). Capability Neurons Localization points out that knowledge cannot be localized, with unexplainability, but capabilities can be localized (Huang et al., 2025b).

**Localization Method.** Distributed Parameters: Knowledge-sensitive neurons are detected using a gradient attribution method, and after sorting, the Top-K neurons are selected and considered as knowledge neurons (Huang et al., 2024). Parameter Layers: Similar to the causal tracing (Meng et al., 2022b), a clean run that predicts the fact, a corrupted run where the prediction is damaged, and a corrupted-with-restoration run that tests the capability of a single state to restore the prediction (Huang et al., 2025a). Parameter Chains: KC (Yao et al., 2024) believes that individual knowledge is stored on a parameter chain and utilizes the entire parameter chain to recall knowledge.

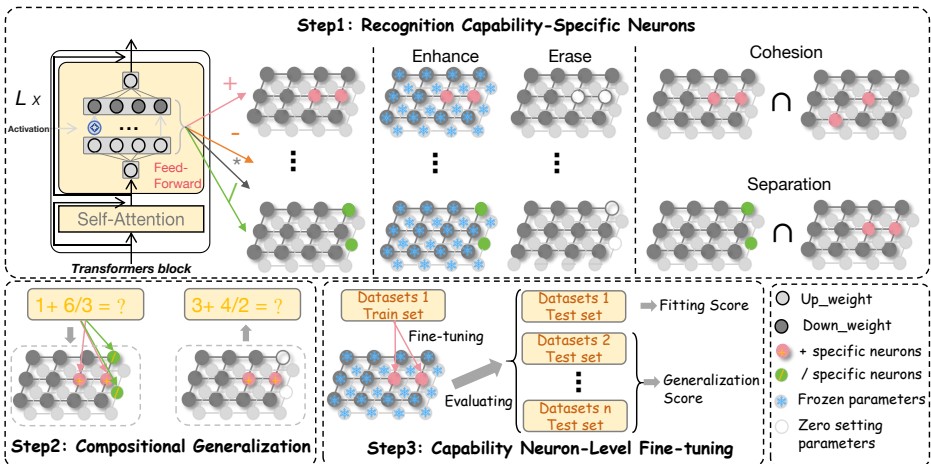

Fig. 1: Illustration of our research methodology. The entire framework consists of three steps: Recognition Capability-Specific Neurons, Compositional Generalization and Capability Neuron-Level Fine-tuning. Step 1 provides identification methods for subsequent research, Step 2 provides interpretability of capability mechanisms for subsequent research, and provides justification for multi-capability enhancement in Step 3.

## 3 METHODOLOGY

Figure 1 illustrates our research approach. First, we proposed a capability neuron detection algorithm and successfully identified capability-specific neurons, which exhibit strong separability and cohesion. Next, we conducted capability compositional generalization experiments and uncovered the mechanism of capability invocation within the model. Finally, based on capability-specific neurons, we proposed a fine-tuning method at the capability neuron level aimed at enhancing multiple capabilities of the model.

### 3.1 CONSTRUCTION OF COMPOSITIONAL GENERALIZATION DATASET

To identify neurons that are highly correlated with specific capabilities utilizing data, it is first necessary to clarify the relationship between capabilities and data (Hinterstoisser et al., 2011). Capabilities reflect the model's proficiency in solving certain problems, they are transferable across datasets, tasks, and even languages. Our compositional generalization dataset consists of two parts: the neuron identification part and the compositional generalization part.

**Neuron Identification Part.** This part helps identify the capability-specific neurons within the model. First, we selected four basic arithmetic operators–"+", "-", "*", and "/", which reflect different computational capabilities. To achieve a cross-task setup, our data includes three types of tasks: Multiple-choice (MQ) tasks, True/False (TF) tasks, and Direct Generation (DG) tasks. Additionally, to meet the cross-language requirement, we utilized Deepseek V3's (Liu et al., 2024a) translation capabilities to provide data in both English and Chinese. The specific data formats are shown in Table 1.

**Combination Generalization Part.** This dataset helps clarify the capability invocation mechanism within the model. The dataset includes five different types of expressions. For example, 2-operator expressions (e.g., 1+3*5), which involve two different operators. Interestingly, to explore the impact of repeatedly occurring operators on neuron activation levels, we designed a 5-operator expression, such as (1+3*5-6/3+5), where the "+" operator appears twice. The specific data formats are shown in Table 1. We provide more dataset details in the Appendix A.

| Multi-operation expression | < + > 
 < 1-OP: 1 + 6 = ?>; | < + ∩ * > 
 < 2-OP:1 + 3*5 = ?>; | ... 
 ..., | < + ∩ - ∩ * ∩ / > 
 < 5-OP: 1+3*5 - 6/3+5=? > |
|---|---|---|---|---|
| **Category** | \multicolumn{2}{c}{**Component Prompt**} | | | **Target Answer** |
| **DG** | Based on the given expression "1+3*5=?", please provide the answer directly: | | | 16 |
| **MQ** | Based on the given expression "1+3*5=?", the following options are provided: (a) 10, (b) 0, (c) 16, and (d) 26. Please select the correct option: | | | c |
| **TF** | Based on the given expression "1+3*5=16", please determine whether it is correct: | | | Yes |

Table 1: Example of combining generalized datasets. The categories are: Directly Generated(DG), Multiple-choice Questions(MQ), and True/False questions(TF). The target answer are "16", "c" and "Yes", respectively. $i$-OP indicates that the operation expression contains $i$ types of operations.

## 3.2 RECOGNITION CAPABILITY-SPECIFIC NEURONS IN LLMS

To identify capability-specific neurons, it is necessary to determine the correlation between each neuron and a specific capability. Previous studies have focused on locating neurons responsible for individual pieces of knowledge (Meng et al., 2022a) or tasks (Leng & Xiong, 2025). Inspired by these works, we propose a Detecting Capability-Specific Neurons (DCSN) method, which utilizes the contribution of each neuron to a capability as a relevance score. Notably, current LLMs are based on the autoregressive transformer architecture, which consists of Multi-Head Self-Attention (MHSA) and Feed-Forward Networks (FFNs) (Touvron et al., 2023). Previous research has demonstrated that FFNs can store a large amount of parametric knowledge (Dai et al., 2021). Our experiments focus solely on this component.

For identifying neuronal samples $S = [s_1, s_2, ..., s_j]$, we compute correlation score for each neuron $n_i$ at layer $l$ as:

$$C(i, l, t, S) = \frac{1}{n} \sum_{j=0}^{n} A_{i,-1}^{l,j} \cdot (W_{un} W_{ud}^l)_{t,i} \tag{1}$$

where $(\cdot)_{t,i}$ represents the $t$-th row and $i$-th column of the input matrix, and $A_{i,-1}^{l,j}$ is the activation output at the last token for neuron $n_i$ at layer $l$ of sample $s_j$. The $W_{un}$ is the unembedding matrix, and $W_{ud}^l$ is the up or down weight of the forward feedback network at layer $l$.

Here we regard $W_{un} W_{ud}^l \in \mathbb{R}^{v \times d_m}$ as a projection function projecting from activations of the neurons to distribution of the vocabulary, where $v$ is the vocabulary and $d_m$ is the intermediate, and regard $A_{i,-1}^l$ as a coefficient of the projection, respectively. This projection clearly displays the average contribution level of each neuron to all samples.

To identify the capability-specific neurons, we take the $Mask$ matrix:

$$Mask_{i,l} = \begin{cases} 1 & |C - mean(C)| > \sigma \cdot var(C) \\ 0 & else \end{cases} \tag{2}$$

where $mean(\cdot)$ denotes the mean value of all scores and $var(\cdot)$ indicates the variance of the neurons. $\sigma$ is the threshold guiding us to find the task neurons. According to statistical principles (Kumar et al., 2007), in the absence of any special instructions to follow, we view the neurons with scores outside $\sigma = 6$ as capability-specific neurons. We also provide experimental results for other $\sigma$ values in the Appendix D.

## 3.3 COMPOSITIONAL GENERALIZATION EXPERIMENT

Once it is confirmed that capability-specific neurons truly exist, we analyze how the model invokes various capabilities when solving problems. First, we design a forward reasoning experiment to observe whether the corresponding neurons are successfully activated when the model solves multi-operation expression problems. Additionally, in the reverse validation experiment, we erase certain

capability-specific neurons in advance to evaluate the model's performance on multi-operation expression problems. Through this process, we aim to clarify the model's internal mechanism for utilizing capabilities, thereby revealing the relationship between capability-specific neurons and model performance.

**Forward Reasoning Experiment.**    We utilize the compositional generalization dataset introduced in Section 3.1 to have the model solve multi-operation expression problems. By analyzing the activation states of neurons in the model, we determine whether capability-specific neurons exhibit compositional generalization. Specifically, we examine whether the activated neurons include the identified capability-specific neurons and whether these neurons correspond to the operators in the multi-operation expressions. This experiment aims to verify whether specific neurons demonstrate compositional generalization properties.

**Reverse Validation Experiment.**    We erase certain capability-specific neurons in advance to observe their impact on the corresponding model capabilities. For example, we set addition-specific neurons to zero and then have the model answer multiple questions containing addition operators, measuring the change in accuracy before and after the erasure. Furthermore, we erase multiple capability-specific neurons simultaneously to examine the resulting accuracy changes in related problems. This step aims to validate the authenticity of the compositional generalization properties of capability-specific neurons.

### 3.4 Enhancing Multiple Capabilities of LLMs Utilizing Capability-Specific Neurons

Through the analysis of capability-specific neurons, we found that these neurons exhibit compositional generalization. This discovery reveals the mechanism by which the model invokes its capabilities and establishes a correspondence between capability-specific neurons and model performance. However, enhancing the model's performance remains a challenge. We propose a Capability Neuron-Level Fine-tuning method (CNLF), aiming to leverage the detected neurons to further improve the model's performance.

First, given a set of training samples $D$, we fine-tune only the corresponding capability-specific neurons during the fine-tuning phase while freezing all other parameters. During the testing phase, inference proceeds as usual. We refer to this approach as Capability Neuron-Level Fine-tuning method. Specifically, existing open-source datasets do not solely focus on a single capability. For instance, the dataset $meta\_math$ (Yu et al., 2023) reflects both the model's mathematical and language capabilities, which poses challenges in selecting which neurons to fine-tune. Fortunately, we have already confirmed the existence of compositional generalization in capabilities. The model's performance on $meta\_math$ is primarily related to its mathematical and language capabilities. Therefore, we choose to fine-tune both the mathematics and language capability-specific neurons simultaneously. Experimental results validate the effectiveness of this method.

## 4 Experiments: Recognition Capability-Specific Neurons

In this section, we first utilized neuron DCSN in Section 3.2 to detect capability-specific neurons, and designed enhancement and erasure experiments to verify the high correlation between these neurons and capabilities. Finally, we constructed dissociation and cohesion indicators to determine the true existence of capability-specific neurons.

### 4.1 Experimental Setup

The first experiment is an enhancement experiment, in which we fine-tune the ability-specific neurons. Specifically, only ability specific neurons are updated, while other parameters in the model are frozen. The second experiment is an inhibition test. During the inference phase, we set the ability specific neurons to zero while keeping other parameters unchanged.

We tested three publicly available models of different sizes, including LLaMA-2-7B (Touvron et al., 2023), LLaMA-3-8B, LLaMA-3-13B and GPT-J-6B (Wang & Komatsuzaki, 2021). The proportion

| Enhancement | LLaMA-2-7B | | | | | GPT-J-6B | | | | |
|---|---|---|---|---|---|---|---|---|---|---|
| | + | - | * | / | Avg. (↑) | + | - | * | / | Avg. (↑) |
| Original | 52.4 | 50.3 | 45.2 | 43.6 | 47.8 | 50.7 | 50.1 | 45.2 | 40.3 | 46.5 |
| FT-Random | 52.7 | 51.0 | 45.7 | 43.9 | 48.3 | 50.9 | 50.7 | 45.3 | 40.5 | 46.8 |
| FT-w/o Cap | 56.7 | 55.6 | 50.6 | 48.3 | 52.8 | 56.3 | 56.2 | 49.3 | 47.6 | 52.3 |
| FT-Cap (Ours) | **67.8** | **68.3** | **66.1** | **65.7** | **66.9** (↑ 14.1) | **66.4** | **67.4** | **65.4** | **65.3** | **66.1** (↑ 13.8) |
| **Erasure** | LLaMA-3-8B | | | | | LLaMA-3-13B | | | | |
| | + | - | * | / | Avg. (↓) | + | - | * | / | Avg. (↓) |
| Original | 56.4 | 55.3 | 48.4 | 48.6 | 52.1 | 58.6 | 57.4 | 50.1 | 51.7 | 54.4 |
| Deactivate-Random | 56.3 | 55.0 | 48.4 | 48.5 | 52.0 | 58.5 | 57.3 | 49.8 | 53.1 | 54.6 |
| Deactivate-Cap (Ours) | **32.5** | **31.5** | **30.4** | **30.8** | **31.3** (↓ 20.7) | **34.6** | **32.7** | **31.6** | **32.7** | **32.9** (↓ 21.7) |

Table 2: Results of enhancement and erasure experiments. FT-Random refers to fine-tuning an equal amount of random parameters (0.05% of parameters), FT-w/o Cap refers to fine-tuning all parameters except for the specified capability-specific neurons (99.95% of parameters), and FT-Cap refers to fine-tuning the specified capability-specific neurons (0.05% of parameters). The experiment only disabled 10% of ability specific neurons.

of ability specific neurons in the total parameters is 0.05%. The datasets is neuron identification part in Section 3.1. The Appendix C provides comparison results with the positioning method NeFT (Xu et al., 2025).

## 4.2 RESULTS

Table 2 presents the results of the enhancement experiments. Compared to fine-tuning an equivalent number of random parameters, fine-tuning capability-specific neurons achieves significant performance improvements. Specifically, despite fine-tuning only 0.05% of the parameters, fine-tuning capability-specific neurons results in a **19.9%** performance improvement. In comparison to "FT-w/o Capability," the performance improvement is 14.1%. Notably, "FT-w/o Capability" refers to fine-tuning all parameters except the capability-specific neurons, which accounts for 99.95% of the total parameters, yet only achieves a mere 4.5% performance improvement. This demonstrates that the identified neurons are highly correlated with the model's capabilities.

Table 2 also provides the results of the erasure experiments. By zeroing out certain neurons and observing the performance drop, the more significant the drop, the more sensitive the erased neurons are to the capability. Despite disabling only 10% of the capability-specific neurons, the model experiences a 21.7% performance drop. When disabling an equivalent number of random parameters (0.05%), the model's performance drops by only 0.1%. This further highlights that the identified neurons are highly sensitive to the model's capabilities.

Meanwhile, previous studies lacked metrics to evaluate the accuracy of neuron identification. Inspired by clustering analysis (MacQueen, 1967), capability-specific neurons for different capabilities should exhibit high distinctiveness (e.g. $n_+$ and $n_-$). The greater the distinctiveness, the higher the identification accuracy. We define this metric as separability $Sep$. Additionally, neurons identified for the same capability (utilizing different datasets, e.g., English $n_{EN}$ and Chinese $n_{CH}$) should exhibit high similarity. The greater the similarity, the stronger the identification reliability. We define this metric as cohesiveness $Coh$. Formally, these metrics can be expressed as:

$$Sep(+ \cap -) = \frac{n_+ \cap n_-}{max(n_+ \cap n_-)}, Coh(+) = \frac{n_{CH} \cap n_{EN}}{max(n_{CH} \cap n_{EN})} \quad (3)$$

Table 3 presents the results of cohesiveness and separability. In previous studies, the cohesiveness of KN (Dai et al., 2021) was only 37.3%, while the separability of ROME (Meng et al., 2022a) reached as high as 86.6%, making the identified neurons unconvincing (Huang et al., 2025b). Even under interference from multiple question types and multilingual data, the capability-specific neurons identified in our study achieve a cohesiveness of **94.3%** and a separability of only **3.6%**. This demonstrates that the identified capability-specific neurons are both reliable and accurate.

In summary, compared to random parameters and "w/o Capability," enhancing and suppressing capability-specific neurons significantly impacts model performance. The results of cohesiveness and

| Cohesiveness | LLaMA-2-7B | | | | | GPT-J-6B | | | | |
|---|---|---|---|---|---|---|---|---|---|---|
| | + | - | * | / | Avg. (↑) | + | - | * | / | Avg. (↑) |
| DG | 95.47 | 93.73 | 92.96 | 94.35 | **94.19** | 95.63 | 93.78 | 93.86 | 94.25 | **94.35** |
| TF | 94.63 | 93.47 | 94.54 | 94.32 | **94.26** | 93.59 | 93.70 | 94.23 | 92.41 | **93.42** |
| MQ | 95.38 | 93.24 | 93.57 | 94.66 | **94.13** | 93.45 | 93.68 | 94.39 | 93.34 | **93.62** |
| CH | 84.11 | 81.63 | 82.77 | 83.32 | **82.97** | 83.54 | 80.55 | 80.76 | 81.62 | **81.59** |

| Separability | LLaMA-3-8B | | | | | LLaMA-3-13B | | | | |
|---|---|---|---|---|---|---|---|---|---|---|
| | + ∩ - | + ∩ * | + ∩ / | * ∩ / | Avg. (↓) | + ∩ - | + ∩ * | + ∩ / | * ∩ / | Avg. (↓) |
| DG | 6.46 | 3.26 | 2.63 | 2.31 | **3.65** | 5.76 | 4.61 | 4.33 | 4.22 | **4.73** |
| MQ | 6.22 | 4.77 | 4.16 | 3.89 | **4.73** | 5.84 | 5.52 | 4.36 | 4.03 | **4.90** |
| TF | 5.73 | 5.05 | 4.53 | 4.06 | **4.82** | 5.65 | 5.08 | 4.51 | 3.76 | **4.71** |
| CH | 5.88 | 4.48 | 4.08 | 3.65 | **4.57** | 4.89 | 4.43 | 4.07 | 3.69 | **4.22** |
| EN | 6.38 | 5.47 | 4.33 | 3.45 | **4.84** | 5.79 | 5.66 | 4.35 | 3.57 | **4.76** |

Table 3: The results of cohesiveness and separability experiments. EN refers to all English datasets, while CH refers to translating all English into Chinese datasets.

separability further demonstrate the high accuracy of neuron identification. Therefore, we conclude that capability-specific neurons do indeed exist in LLMs.

## 5 EXPERIMENTS: COMBINATION GENERALIZATION

We analyzed ability specific neurons to understand the ability calling mechanism of the model. According to the analysis method in Section 3.3, we have listed the relevant results of combination generalization experiments.

### 5.1 FORWARD REASONING EXPERIMENT

**Experimental Setup.** The experimental data $D = [d_1, d_2, ..., d_r]$ is the combined generalization part in Section 3.1. Utilizing Equation 2, we identified activated neurons and compared them with the four ability specific neurons obtained in Section 4, utilizing the coincidence ratio as the evalua-

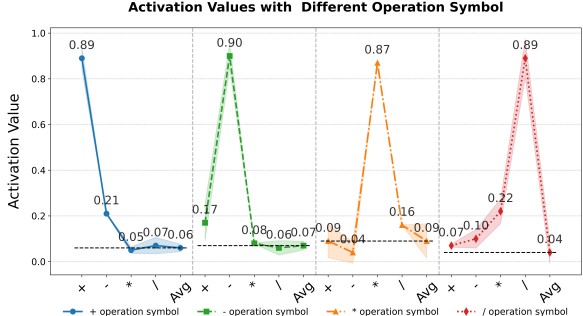

Fig. 2: The activation values corresponding to different operation symbols, and different colors refer to different operators in 1-OP.

tion metric. Specifically, when the experimental data is a 2-operation expression (e.g. "1+3 * 5"), we calculate the ratio $R$ of activated neurons $n_{act}$ containing $\{+\}$-specific neurons $n_+$ and $\{*\}$-specific neurons $n_*$ as the evaluation metric. Formalized as follows:

$$R(+\&*) = \frac{1}{r}\sum_{i=0}^{r}\left(\frac{n^i_{(+|act)}}{n^i_+} + \frac{n^i_{(*|act)}}{n^i_*}\right) \tag{4}$$

where $n^i_{(+|act)}$ refers to activated $\{+\}$-specific neurons, $n^i_+$ refers to all $\{+\}$-specific neurons.

**Results.** Table 4 presents the activation rates of various arithmetic expressions. The results show that, on average, **89.53%** of the capability-specific neurons are activated when the model processes multi-operation expressions. The activation rates for 4-OP and 5-OP are similar, indicating that the presence of two identical operators in an expression does not significantly increase the activation rate of the corresponding capability-specific neurons. Figure 2 further illustrates the activation values of neurons when processing different arithmetic expressions. Counterintuitively, when an expression contains only the "+" operator, the activation value for "-" is also significantly higher than the average. A similar issue is observed with "*" and "/". This contradicts our intuition, as under normal circumstances, when an arithmetic expression does not contain "-", the activation value for

| Forward | LLaMA-2-7B | | | | | GPT-J-6B | | | | |
|---|---|---|---|---|---|---|---|---|---|---|
| | 2-OP | 3-OP | 4-OP | 5-OP | Avg.(↑) | 2-OP | 3-OP | 4-OP | 5-OP | Avg. (↑) |
| DG | 94.37 | 89.67 | 87.43 | 86.66 | **89.53** | 90.63 | 87.64 | 84.13 | 83.22 | **86.41** |
| MQ | 91.22 | 86.42 | 82.67 | 82.64 | **85.73** | 89.76 | 86.42 | 82.37 | 82.66 | **85.30** |
| TF | 90.37 | 87.45 | 83.64 | 83.37 | **86.21** | 89.73 | 87.64 | 82.66 | 82.47 | **85.63** |
| CH | 89.67 | 85.43 | 81.09 | 80.43 | **84.16** | 90.13 | 86.57 | 82.93 | 82.86 | **85.62** |
| EN | 90.17 | 86.59 | 82.19 | 81.93 | **85.22** | 89.30 | 85.71 | 81.05 | 80.54 | **84.15** |
| Reverse | LLaMA-3-8B | | | | | LLaMA-3-13B | | | | |
| | 2-OP | 3-OP | 4-OP | 5-OP | Avg.(↓) | 2-OP | 3-OP | 4-OP | 5-OP | Avg. (↓) |
| Original | 60.25 | 56.47 | 54.29 | 54.17 | 56.29 | 66.54 | 63.41 | 60.37 | 59.29 | 62.40 |
| Deactivate-Random | 60.27 | 56.21 | 54.20 | 53.07 | 55.93 | 66.34 | 63.20 | 60.35 | 59.28 | 62.29 |
| Ours | **32.07** | **30.49** | **27.31** | **27.78** | **29.41** (↓ 26.88) | **34.22** | **31.72** | **28.57** | **27.43** | **30.48** (↓ 29.92) |

Table 4: The experimental results of forward reasoning and reverse verification.

"-" should be close to the average. We analyzed the underlying reason for this phenomenon, which might be related to the fact that addition and subtraction are inverse operations. To investigate further, we conducted ablation experiments.

**Ablation Experiment.** In Table 5, such as in arithmetic expressions where only addition is present, the performance drops significantly by 6.3% when subtraction is removed compared to multiplication and division. This indicates a certain correlation between addition and subtraction, which is related to their nature as inverse operations, thus verifying our analysis.

| Erase | LLaMA-3-8B (↓) | | | |
|---|---|---|---|---|
| | + | - | * | / |
| 1-OP (+) | ↓ **23.9** | ↓ 6.3 | ↓ 0.1 | ↓ 0.2 |
| 1-OP (-) | ↓ 5.7 | ↓ **23.8** | ↓ 0.0 | ↓ 0.1 |
| 1-OP (*) | ↓ 0.0 | ↓ 0.3 | ↓ **18.0** | ↓ 6.0 |
| 1-OP (/) | ↓ 0.4 | ↓ 0.1 | ↓ 5.3 | ↓ **17.8** |

Table 5: The results of the ablation experiment. 1-OP (+) refers to the presence of + in 1-OP. The value refers to the magnitude of the performance decline with zero capability-specific neurons. (↓) represents the influence between neurons that perform inverse operations on each other.

## 5.2 REVERSE VERIFICATION EXPERIMENT

**Experimental Setup.** We sequentially ablate the capability-specific neurons corresponding to operators in expressions to observe their impact on final performance. For 2-OP, we only remove a single capability neuron, whereas for 3-OP and 4-OP, we remove multiple capability neurons. This further verifies the authenticity of capability compositional generalization.

**Results.** In Table 4, for 2-OP, removing the corresponding capability-specific neuron significantly reduces the model's performance by **28.18%**. Notably, for 3-OP and 4-OP, we intuitively observe that as more corresponding capability-specific neurons are removed, the performance degradation becomes more pronounced. However, when removing an equivalent number of parameters, the performance drop is minimal.

In conclusion, we have gained an understanding of the model's capability invocation mechanism during forward inference and believe that the model exhibits compositional generalization of capabilities.

## 6 EXPERIMENT: ENHANCING MULTIPLE CAPABILITIES OF LLMs

To fully leverage capability-specific neurons, we propose a fine-tuning method at the capability-specific neuron level. The experimental results show that, in addition to operation-related capabilities, our work is also applicable to deeper-level capabilities (such as Math, Program, and Language, etc.).

### 6.1 EVALUATION INDICATORS

To evaluate the fitting and generalization capabilities of neurons, we designed two evaluation metrics: the fitting score on the test set, and the generalization score obtained by training on the current dataset and testing on other datasets.

| 2-OP | LLaMA-2-7B/+(↑) | | | | GPT-J-6B/+(↑) | | | |
|---|---|---|---|---|---|---|---|---|
| | + | +&- | +&* | +&/ | + | +&- | +&* | +&/ |
| Original | 52.41 | 47.63 | 43.65 | 41.65 | 50.7 | 47.31 | 43.53 | 39.41 |
| FT-Random | 52.71 | 48.07 | 44.72 | 42.67 | 50.92 | 48.38 | 44.37 | 40.55 |
| O-LoRA | 67.4 | 49.77 | 47.32 | 44.97 | 64.89 | 59.63 | 46.55 | 42.09 |
| FT-All | **68.36** | 49.86 | 47.39 | 45.36 | **66.93** | 59.83 | 47.29 | 43.63 |
| Ours | 67.83 | **57.64**(↑ 7.78 ) | **58.83**(↑ 11.44 ) | **57.42**(↑ 12.06 ) | 66.42 | **66.81**(↑ 6.98) | **58.33**(↑ 11.04) | **53.37**(↑ 9.74 ) |
| **Math** | LLaMA-2-7B/GSM8K(↑) | | | | GPT-J-6B/GSM8K(↑) | | | |
| | GSM8K | Meta_Math | SVAMP | AMC | GSM8K | Meta_Math | SVAMP | AMC |
| Original | 21.42 | 23.63 | 42.55 | 30.01 | 23.04 | 23.07 | 41.09 | 30.89 |
| FT-Random | 21.21 | 23.67 | 41.99 | 30.98 | 24.09 | 23.42 | 42.70 | 31.47 |
| O-LoRA | 35.09 | 24.37 | 40.36 | 31.09 | 37.42 | 26.37 | 45.09 | 30.82 |
| FT-All | **38.67** | 25.63 | 41.32 | 31.72 | **39.09** | 26.63 | 44.92 | 31.78 |
| Ours | 35.02 | **35.03**(↑ 9.40 ) | **50.20**(↑ 8.88 ) | **46.01**(↑ 14.29 ) | 40.27 | **33.42**(↑ 6.79 ) | **54.32**(↑ 9.4 ) | **48.06**(↑ 16.28 ) |
| **Program** | LLaMA-2-7B/Code25K(↑) | | | | GPT-J-6B/Code25K(↑) | | | |
| | Code25K | HumanEval | MBPP | APPS | Code25K | HumanEval | MBPP | APPS |
| Original | 23.51 | 27.99 | 43.51 | 35.42 | 43.03 | 24.07 | 36.08 | 32.82 |
| FT-Random | 24.32 | 28.09 | 41.99 | 36.98 | 43.29 | 24.52 | 37.74 | 33.04 |
| O-LoRA | 34.00 | 36.96 | 47.32 | 41.08 | 48.52 | 29.37 | 43.08 | 38.42 |
| FT-All | **39.47** | 37.02 | 48.53 | 42.29 | **49.89** | 29.98 | 44.62 | 38.87 |
| Ours | 35.74 | **45.37**(↑ 8.35 ) | **58.23**(↑ 9.70 ) | **49.73**(↑ 7.44 ) | 55.73 | **38.92**(↑ 8.94 ) | **56.42**(↑ 11.80 ) | **49.83**(↑ 10.96 ) |
| **Language** | LLaMA-2-7B/Emotion(↑) | | | | GPT-J-6B/Emotion(↑) | | | |
| | Emotion | Imdb | GoEmotions | TweetEval | Emotion | Imdb | GoEmotions | TweetEval |
| Original | 67.42 | 57.43 | 66.78 | 57.43 | 60.14 | 53.01 | 51.92 | 52.82 |
| FT-Random | 68.22 | 58.72 | 67.42 | 58.99 | 62.43 | 54.32 | 53.71 | 54.42 |
| O-LoRA | 76.43 | 57.62 | 68.93 | 58.44 | 73.32 | 56.57 | 51.53 | 51.55 |
| FT-All | **79.85** | 58.66 | 68.79 | 58.79 | **75.08** | 53.32 | 52.42 | 53.78 |
| Ours | 77.47 | **66.70**(↑ 8.04 ) | **77.90**(↑ 9.11 ) | **67.09**(↑ 8.30 ) | 73.06 | **69.43**(↑ 16.11 ) | **63.17**(↑ 10.75 ) | **65.07**(↑ 11.29 ) |

Table 6: Experimental results on ability enhancement and generalization. " LLaMA-2-7B/+ " refers to fine-tuning the "+" training set on {+}-specific neurons of LLaMA-2-7B, and evaluate the fitting and generalization ability on "+" and three other 2-OP datasets.

## 6.2 EXPERIMENTAL SETUP

In this Section, we provide an introduction to datasets and baselines. The dataset includes Math, Program, and Language capabilities, with baselines including FT Random, O-LoRA (Wang et al., 2023), and FT-ALL (Hawthorne & Isaacs, 2018) methods. We provide more details in Appendix B.

## 6.3 THE RESULTS OF THE OPERATION-RELATED CAPABILITIES

Table 6 presents the results of model enhancement on the addition {+}-OP. By fine-tuning the neurons related to the {+}-OP, we evaluate the model's performance on 1-OP(+) and on 2-OP tasks that include {+}-OP. The results show that our method achieves a fitting capability for the {+}-OP comparable to that of O-LoRA. For generalization to other 2-OP operations, our method outperforms FT-ALL by 12.06%. These findings demonstrate that our method possesses superior generalization ability and genuinely improves the model's {+}-capability.

## 6.4 THE RESULTS OF DEEPER-LEVEL CAPABILITIES

Furthermore, we shift our attention to deeper-level capabilities (such as Math, Program, and Language), which hold significant practical value. Table 6 shows that by fine-tuning only 0.05% of the parameters, we can achieve fitting scores comparable to FT-All and O-LoRA. Meanwhile, our generalization scores are **15.89%** higher than FT-All. Compared to fine-tuning the same number of random parameters, our performance has improved by **18.17%**. The experimental results demonstrate that we provide a cost-effective fine-tuning approach and exhibit significant generalization capabilities on other capabilities. This Discussion E provides more insights into our research.

## 7 CONCLUSION

In this study, we propose a methodological research framework to understand the capability mechanisms of models from a neuronal perspective. First, we identified that capability-specific neurons are indeed present. Second, we gained an initial understanding of the internal capability invocation mechanisms within models. Finally, we introduced a low-cost, high-generalization fine-tuning paradigm that leverages capability-specific neurons to enhance various model capabilities. Importantly, this research framework promotes advancements in model interpretability studies.

## ETHICS STATEMENT

This work is the first to discover the compositional generalization phenomenon of capability neurons and proposes a low-cost, high-generalization neuron-level fine-tuning method. All experiments are based on publicly available datasets and open-source models, with no involvement of human subjects or private data. This benchmark is intended for academic research on model compression rather than for harmful applications. We have not identified significant ethical risks related to bias, privacy, or abuse. All experiments comply with the license terms of the datasets and models used.

## REPRODUCIBILITY STATEMENT

We provide detailed descriptions of the benchmark construction, evaluation protocols, and experimental setup. All underlying datasets are publicly available, and we followed standard preprocessing and evaluation procedures. Additional details and complete results are reported in the appendix.

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

## A  THE CONSTRUCTION PROCESS AND NUMBER OF SAMPLES IN THE DATASET

Construction Process: Overall, the neuron recognition part only contains 1-OP, while the combinatorial generalization part includes 2,3,4,5-OP. Specifically, as shown in Table 7, we use the same prompt framework and only change different arithmetic expressions. For example, for Direct Generation (DG), the prompt for sample 1 is "Based on the given expression 1+3*5=?, please provide the answer directly:", and for sample 2, we only need to change "1+3*5=?" to "2+6/4=?". Regarding the arithmetic expressions, the numbers are randomly generated while ensuring an equal distribution of +, -, *, and / operators.

The number of examples is shown as follows:

| English | 1-OP | 2-OP | 3-OP | 4-OP | 5-OP |
|---------|------|------|------|------|------|
| DG | 4000 | 1000 | 1000 | 1000 | 1000 |
| MQ | 4000 | 1000 | 1000 | 1000 | 1000 |
| TF | 4000 | 1000 | 1000 | 1000 | 1000 |

Table 7: The number of samples in the dataset.

The number of Chinese samples is the same as English ones, which were translated using Deepseek V3 (Liu et al., 2024a). In 1-OP (4000 samples), there are 1000 samples each for +, -, *, and /. We will add this explanation in the revised version and commit to making all data publicly available.

## B  EXPERIMENTAL SETUP

### B.1  DATASET AND NEURON DETECTION

We collected datasets related to mathematics, programming, and language, and used the method described in Section 3.1 to identify math-specific neurons, programming-specific neurons, and language-specific neurons. The relevant datasets include:

- **Math:** GSM8K (Cobbe et al., 2021) contains approximately 8,000 elementary math problems with detailed solutions. Meta_Math (Yu et al., 2023) focused on meta-learning for math problems. MathQA (Luo & Pan, 2024), SVAMP (Naeem et al., 2014) and AMC (AMCs, 2013) datasets.
- **Program:** Code25K (Beguš, 2021) contains around 25,000 code snippets. HumanEval (Zheng et al., 2023), MBPP (Athiwaratkun et al., 2022), CodeXGLUE (Lu et al., 2021) and APPS (Zheng et al., 2023).
- **Language:** Emotion (Kosti et al., 2019) with text data labeled with various emotions. Imdb (Tripathi et al., 2020) contains movie reviews and ratings. GoEmotions (Demszky et al., 2020), SemEval-2019 Task 3 (Chatterjee et al., 2019) and TweetEval (Barbieri et al., 2020) .

### B.2  BASELINES

The baseline methods mainly include:

- **FT-Random:** Fine-tuning an equal amount of random parameters.
- **O-LoRA** (Wang et al., 2023) : learning tasks in different (low-rank) vector subspaces.
- **FT-All** (Hawthorne & Isaacs, 2018): Fine-tuning the entire model using training data.

## C  THE COMPARATIVE EXPERIMENTAL RESULTS

We provide comparison results with existing positioning methods. The evaluation indicators include Cohesiveness and Separability.

In Figure 8, the results show that our proposed method exhibits characteristics such as high Cohesiveness and low Separation, demonstrating superior performance.

| DG | Cohesiveness(↑) | Separability(↓) |
|---|---|---|
| NeFT | 72.33 | 35.89 |
| Ours | **94.19** | **3.65** |
| **MQ** | **Cohesiveness(↑)** | **Separability(↓)** |
| NeFT | 70.59 | 32.68 |
| Ours | **94.26** | **4.82** |
| **TF** | **Cohesiveness(↑)** | **Separability(↓)** |
| NeFT | 73.17 | 33.83 |
| Ours | **94.13** | **4.73** |

Table 8: Comparison of our method and NeFT's (Xu et al., 2025) localization results.

# D  SUPPLEMENTARY EXPERIMENTAL RESULTS

## D.1  SUPPLEMENTARY EXPERIMENTS

| Enhancement | LLaMA-2-7B | | | | | GPT-J-6B | | | | |
|---|---|---|---|---|---|---|---|---|---|---|
| | + | - | * | / | Avg. (↑) | + | - | * | / | Avg. (↑) |
| Original | 52.4 | 50.3 | 45.2 | 43.6 | 47.8 | 50.7 | 50.1 | 45.2 | 40.3 | 46.5 |
| FT-Random | 52.7 | 51.0 | 45.7 | 43.9 | 48.3 | 50.9 | 50.7 | 45.3 | 40.5 | 46.8 |
| FT-w/o Cap | 56.7 | 55.6 | 50.6 | 48.3 | 52.8 | 56.3 | 56.2 | 49.3 | 47.6 | 52.3 |
| FT-Cap (Ours) | **67.4** | **68.0** | **65.1** | **63.1** | **62.1** | **65.1** | **66.4** | **64.3** | **64.2** | **62.5** |
| **Erasure** | **LLaMA-3-8B** | | | | | **LLaMA-3-13B** | | | | |
| | + | - | * | / | Avg. (↓) | + | - | * | / | Avg. (↓) |
| Original | 56.4 | 55.3 | 48.4 | 48.6 | 52.1 | 58.6 | 57.4 | 50.1 | 51.7 | 54.4 |
| Deactivate-Random | 56.3 | 55.0 | 48.4 | 48.5 | 52.0 | 58.5 | 57.3 | 49.8 | 53.1 | 54.6 |
| Deactivate-Cap (Ours) | **34.6** | **33.7** | **32.2** | **31.7** | **32.7** | **36.1** | **34.0** | **32.5** | **33.7** | **33.1** |

Table 9: Results of enhancement and erasure experiments. $\sigma = 3$.

| Math | LLaMA-3-8B/GSM8K(↑) | | | | LLaMA-3-13B/GSM8K(↑) | | | |
|---|---|---|---|---|---|---|---|---|
| | GSM8K | Meta_Math | SVAMP | AMC | GSM8K | Meta_Math | SVAMP | AMC |
| Original | 24.32 | 26.73 | 43.67 | 32.17 | 26.15 | 27.39 | 45.73 | 33.67 |
| FT-Random | 25.31 | 25.43 | 43.39 | 31.74 | 27.33 | 27.41 | 45.73 | 33.59 |
| O-LoRA | 38.94 | 27.33 | 44.63 | 35.97 | 41.32 | 29.45 | 47.12 | 34.63 |
| FT-All | **43.07** | 28.53 | 45.32 | 35.67 | **43.89** | 29.63 | 47.82 | 35.87 |
| Ours | 37.42 | **36.73**(↑ 8.20 ) | **57.92**(↑ 12.20) | **47.63**(↑ 11.96) | 41.23 | **36.47**(↑ 6.84 ) | **58.63**(↑ 10.81) | **51.76**(↑ 15.89) |

Table 10: Experimental results on ability enhancement and generalization. Fine-tuning the GSM8K training set and evaluate the fitting and generalization ability on GSM8K and three other datasets.

| Cohesiveness | LLaMA-3-8B | | | | | LLaMA-3-13B | | | | |
|---|---|---|---|---|---|---|---|---|---|---|
| | + | - | * | / | Avg. (↑) | + | - | * | / | Avg. (↑) |
| DG | 94.32 | 92.97 | 93.67 | 93.35 | **93.29** | 94.62 | 93.64 | 93.56 | 94.37 | **93.35** |
| TF | 94.33 | 93.27 | 93.64 | 94.02 | **93.36** | 93.49 | 93.67 | 93.29 | 92.51 | **93.78** |
| MQ | 95.42 | 93.27 | 92.59 | 92.67 | **93.23** | 93.55 | 93.69 | 93.32 | 93.42 | **93.22** |
| CH | 83.61 | 82.91 | 83.74 | 85.42 | **83.27** | 84.24 | 81.54 | 82.46 | 81.55 | **82.79** |
| **Separability** | **LLaMA-2-7B** | | | | | **GPT-J-6B** | | | | |
| | + ∩ - | + ∩ * | + ∩ / | * ∩ / | Avg. (↓) | + ∩ - | + ∩ * | + ∩ / | * ∩ / | Avg. (↓) |
| DG | 5.64 | 4.27 | 3.93 | 2.51 | **3.45** | 5.66 | 4.31 | 4.00 | 3.78 | **4.23** |
| MQ | 5.32 | 4.65 | 4.23 | 3.79 | **4.03** | 5.88 | 5.42 | 4.74 | 4.12 | **4.98** |
| TF | 5.63 | 5.15 | 4.70 | 4.02 | **4.61** | 5.60 | 5.18 | 4.52 | 3.74 | **4.73** |
| CH | 5.76 | 4.40 | 4.12 | 3.75 | **4.24** | 5.78 | 4.52 | 4.01 | 3.62 | **4.03** |
| EN | 6.24 | 5.41 | 4.09 | 3.66 | **4.38** | 5.40 | 4.58 | 4.30 | 3.46 | **4.68** |

Table 11: The results of cohesiveness and separability experiments. EN refers to all English datasets, while CH refers to translating all English into Chinese datasets.

## D.2 ABLATION EXPERIMENT OF HYPERPARAMETRIC $\sigma$

We provide additional ablation experiments, which are conducted on Llama-2-7B. The values in the table are $Avg.$ metrics, and the experimental settings are the same as those in Table 2.

| Enhancement Experiment | | | | | | | | | |
|---|---|---|---|---|---|---|---|---|---|
| $\sigma$ | 1 | 2 | 3 | 4 | 5 | 6 | 7 | 8 | 9 |
| Avg. ($\uparrow$) | 54.5 | 57.2 | 62.1 | 63.4 | 65.5 | **66.9** | 65.4 | 62.7 | 60.8 |
| Erasure Experiment | | | | | | | | | |
| $\sigma$ | 1 | 2 | 3 | 4 | 5 | 6 | 7 | 8 | 9 |
| Avg. ($\downarrow$) | 44.3 | 38.9 | 35.4 | 30.8 | 25.5 | **22.6** | 24.7 | 29.9 | 34.1 |

Table 12: The Ablation Experiment Results of Hyperparametric $\sigma$.

Experimental results show that $\sigma = 6$ achieves superior performance, which demonstrates the rationality of our setting $\sigma = 6$.

## D.3 THE RESULTS OF LARGER SCALE MODELS

We conducted enhancement and erasure experiments on the LLaMA-2-30B and LLaMA-2-70B models. The remaining experimental settings are the same as those in the paper. The results are as follows:

| LLaMA-2-30B | + | - | * | / | Avg. ($\uparrow$) |
|---|---|---|---|---|---|
| Original | 54.7 | 51.6 | 46.3 | 44.8 | 49.3 |
| FT-Random | 53.4 | 52.1 | 46.7 | 45.3 | 49.4 |
| FT-Cap (Ours) | 68.9 | 69.4 | 67.5 | 66.8 | 68.1 |
| **LLaMA-2-70B** | + | - | * | / | Avg. ($\uparrow$) |
| Original | 55.4 | 52.6 | 47.3 | 45.1 | 50.1 |
| FT-Random | 54.3 | 52.9 | 47.4 | 46.5 | 50.3 |
| FT-Cap (Ours) | 69.3 | 69.9 | 68.4 | 67.5 | 68.8 |

Table 13: Enhancement Experiment.

| LLaMA-2-30B | + | - | * | / | Avg. ($\downarrow$) |
|---|---|---|---|---|---|
| Original | 54.7 | 51.6 | 46.3 | 44.8 | 49.3 |
| Deactivate-Random | 54.4 | 51.1 | 46.1 | 44.4 | 49.0 |
| Deactivate-Cap (Ours) | 34.5 | 32.6 | 33.8 | 32.7 | 33.4 |
| **LLaMA-2-70B** | + | - | * | / | Avg. ($\downarrow$) |
| Original | 55.4 | 52.6 | 47.3 | 45.1 | 50.1 |
| Deactivate-Random | 55.2 | 52.3 | 47.0 | 46.9 | 50.4 |
| Deactivate-Cap (Ours) | 35.1 | 33.2 | 34.3 | 33.2 | 33.9 |

Table 14: Erasure Experiment.

Experimental results show that our method still has significant performance advantages when extended to larger-scale models.

## E APPENDIX: DISCUSSION

### E.1 NOT ONLY IDENTIFIED THE {+,-,*, / } CAPABILITY-SPECIFIC NEURONS

One of the core contributions of our work is to demonstrate that capability neurons have combinatorial generalization. We first demonstrated this intriguing phenomenon using four operators. In addition to

identifying capability-specific neurons for four operators, we identified capability-specific neurons for mathematics, language, and programming in Section 3.4 and Experiment 6.

However, as we mentioned, the existing datasets do not have as good capability differentiation as the four operators we constructed when identifying capability-specific neurons for mathematics, language, and programming. For example, the meta_math dataset, which contains both mathematical and language capabilities, fine-tuning some neurons would disrupt other abilities of the model (Huang et al., 2025b), which is the biggest challenge faced by previous methods for practical application.

Fortunately, by leveraging the phenomenon of combinatorial generalization of capability neurons we discovered, and fine-tuning the capability neurons we located, we not only achieved significant performance improvements but also good generalization (Table 6). The discovery of the combinatorial generalization phenomenon will broadly aid future neuron fine-tuning tasks.

**This explains the confusion in previous work:** Adjusting the neurons of a certain knowledge or task would lead to a decrease in performance in other knowledge or tasks and poor generalization. This is because their premise (Leng & Xiong, 2025; Meng et al., 2022b) (that knowledge or tasks can be parameter localized) is incorrect.

### E.2 Regarding Other Works Exploring Neuron-Related Capabilities and Compositional Generalization

Firstly, aggregation and separation are used as accuracy indicators for neuron localization. Based on cluster analysis, we are the first to propose these indicators for cross-comparing different localization methods. The results show that it is unreasonable to utilize knowledge or tasks as localization objects, proving that the previous assumptions of parameter localization have serious flaws. We have experimentally proven that it is accurate to utilize abilities as localization objects. Methods (Meng et al., 2022a; Leng & Xiong, 2025) that utilize knowledge or tasks as neuron localization objects cannot predict the experimental conclusions in our work, and these methods has not explicitly discovered the phenomenon of combinatorial generalization.

Moreover, the main contribution of our work is the first discovery that capability-specific neurons have the phenomenon of combinatorial generalization. Specifically, using four operators (=, -, *, /), we determined the capability call mechanism within the model. When different capability-specific neurons are activated in combination, complex tasks (such as 3-OP) can be accurately solved. When capability-specific neurons are enhanced, the model can generalize well across datasets and tasks (Table 6). This explains the confusion in previous work: enhancing neurons for a specific knowledge or task leads to a decline in the performance of other knowledge or tasks and poor generalization. This is because their preliminary assumption (knowledge or tasks can be parameter localized) is wrong.

Finally, we discovered the phenomenon of combinatorial generalization using four operators, but the conclusion is not limited to mathematical capability. We identified math, language, and programming capability-specific neurons in Section 3.4 and Experiment 6. However, as we mentioned, when identifying math, language, and programming capability-specific neurons, existing datasets do not have good capability differentiation like our constructed four operators. For example, the meta_math dataset, which contains both mathematical and language capabilities, fine-tuning some neurons will disrupt other capabilities of the model (Huang et al., 2025a), which is the biggest challenge faced by previous methods in practical implementation. Fortunately, by utilizing the phenomenon of combinatorial generalization of capability neurons we discovered, and fine-tuning the capability neurons we located, not only is there a significant performance improvement, but also good generalization (Table 6). The discovery of the combinatorial generalization phenomenon will broadly assist future neuron fine-tuning tasks. This finding has high novelty and practicality, which was not proposed in previous work, and our experimental conclusions could not be accurately predicted.

### E.3 Distinguishing between "Capability" and "Task"

First, from a formal perspective, a task merely refers to having the model complete a specific piece of work, such as a question-answering or named entity recognition task. In this process, the model will utilize various abilities, such as language and reasoning. However, ability is an inherent attribute of the model itself; it is not limited to the form of a task and exists across tasks. The experimental results

in Table 6 show that enhancing the model's abilities can help improve its performance on current tasks and even unseen tasks, exhibiting significant generalization, which task neurons cannot achieve.

From the perspective of localization, the localization of task neurons relies on specific task datasets, such as question-answering datasets (MQA, ARC). In contrast, the localization of ability neurons depends on datasets that reflect specific abilities, such as mathematical ability datasets (GSM8K, Meta_Math). Ability is the product of aggregated localization from multiple related datasets, which makes it more reasonable and accurate. This is verified by the experiments in Table 4 and Table 6.

## F    LIMITATIONS

Due to limitations in computing resources, we did not conduct relevant experiments on larger language models. Due to the novelty of our method, existing methods do not have the same experimental setup and lack comparisons with more datasets and localization methods.

