# OpenReview forum: "Combination Generalization of Capability-Specific Neurons in LLMs"
_ICLR.cc/2026/Conference — Submitted to ICLR 2026_

### Official Review · Reviewer_3GuE · 2025-10-22

**Soundness:** 3
**Presentation:** 3
**Contribution:** 2
**Rating:** 2
**Confidence:** 5

**Summary:**

This paper proposes a method for detecting capability-specific neurons in LLMs and claims to discover that these neurons exhibit compositional generalization. The authors construct a dataset based on arithmetic operators (+, -, *, /), propose a Detecting Capability-Specific Neurons (DCSN) method, and introduce a Capability Neuron-Level Fine-tuning (CNLF) approach.

**Strengths:**

The paper is well-organized with a logical flow from neuron detection to compositional generalization to fine-tuning applications.

The introduction of cohesion and separation metrics provides a principled way to evaluate neuron localization quality, which is valuable for comparing different approaches.

The observation that capability-specific neurons exhibit compositional generalization (e.g., addition and multiplication neurons both activate for "1+3*5") is noteworthy.

**Weaknesses:**

There is absolutely no technical novelty in the way neurons are selected. NeFT (COLING 2025) and Robustness-Preserving Fine-tuning (ECCV 2024) show that the finetuning method is also not a novel parameter efficient finetuning method

The domain in which the results are evaluated is very narrow (four arithmetic operators), far below ICLR standard

The mechanistic interpretability literature is crowded with similar work:
Knowledge neurons (Dai et al., 2021)
Task neurons (Leng & Xiong, 2025)
Causal tracing (Meng et al., 2022)

**Questions:**

No additional questions

---

> ### Author Response · Authors · 2025-11-17
>
> Thank you for your feedbacks on the paper!
>
> **Q1:** The selection method of neurons lacks any technical innovation.
>
> W1: First, our DCSN is a localization method based on capability-specific neurons, which has not been proposed in previous works. Additionally, the core contributions and innovations of this paper lie in revealing the capability invocation mechanism inside the model and discovering the combinatorial generalization phenomenon of capability-specific neurons, which are of great significance for the field of interpretability. Furthermore, we elaborate on the innovations of the paper in detail:
> - **Indicator Innovation**: We are the first to adopt aggregation and separation as accuracy indicators for neuron localization and apply them to cross-comparison of different localization methods. This innovation reveals that previous methods [1,2] targeting knowledge or tasks for localization are seriously flawed, and proves the rationality of using abilities as localization objects. Methods that take knowledge or tasks as localization objects cannot predict our experimental conclusions and have not discovered the phenomenon of combinatorial generalization.
> - **Phenomenon Discovery**: We first discovered the phenomenon of combinatorial generalization in capability-specific neurons. By using four operators (+, -, *, /), we determined the capability call mechanism within the model. When different capability-specific neurons are activated in combination, complex tasks (such as 3-OP) can be accurately solved. When capability-specific neurons are enhanced, the model can generalize well across datasets and tasks (Table 6). This explains the confusion in previous work [2]: enhancing neurons for specific knowledge or tasks leads to a decline in the performance of other knowledge or tasks and poor generalization, because their preliminary assumption (that knowledge or tasks can be parameter-localized) is wrong.
> - **Method Universality**: We discovered the phenomenon of combinatorial generalization using four operators, and this conclusion is not limited to mathematical capabilities. We also identified capability-specific neurons for language and programming in Section 3.4 and Experiment 6. However, existing datasets are insufficient in distinguishing capabilities, while our method can effectively address this challenge. By leveraging the phenomenon of combinatorial generalization of capability neurons for fine-tuning, we can not only significantly improve model performance but also achieve good generalization (Table 6). The discovery of the phenomenon of combinatorial generalization will broadly assist future neuron fine-tuning tasks. This finding is highly novel and practical, which has not been proposed in previous work, and our experimental conclusions cannot be accurately predicted.
>
> ---
>
> **Q2:** The domain in which the results are evaluated is very narrow (four arithmetic operators).
>
> **W2:** The reviewer may have overlooked the content on page 8 and beyond of the paper. In addition to the operator experiments mentioned by the reviewer, we have also conducted experiments on higher-level capabilities, and the results demonstrate the rationality of our capability-specific neurons. First, in Section 6, we performed experiments on three types of capabilities—mathematics, programming, and language. The results show that capability neurons exhibit excellent separability and aggregation, and our fine-tuning method for capability neurons has significant generalization. Furthermore, in Appendix E, we offer additional discussions beyond arithmetic capabilities, fully elaborating on the paper’s logic from arithmetic operators to diverse capabilities, and providing a detailed description of the definition of model capabilities.
>
> ---
>
> **Q3:** The mechanistic interpretability literature is crowded with similar work.
>
> **W3:** Thank you for the reviewer’s unique insights into this field. As the three papers provided by the reviewer indicate, they represent three previous knowledge localization hypotheses: scattered parameters, parameter layers, and parameter chains. However, Paper [3] has proven the unreasonableness of these three hypotheses, which is also one of the motivations for our paper. Through experiments, we have demonstrated that model capabilities have significant advantages in parameter localization and are the first to reveal the capability invocation mechanism inside the model. These contributions have not been achieved in previous works and have important implications for the field of interpretability. We confirm that our paper has no similarities with previous studies except that we all conduct research in the field of parameter localization.
>
> **References**
>
> [1] Towards understanding multi-task learning (generalization) of llms and exploring task-specific neurons.
>
> [2] Editing models with task arithmetic.
>
> [3] Capability Localization: Capabilities Can be Localized rather than Individual Knowledge.

---

> > ### Comment · Reviewer_3GuE · 2025-11-17
> > **Thank you for this lesson in modesty**
> >
> > I would like to apologize to the authors for my misunderstanding of Table 6, which led me to discount this work. I've edited my review (the modifications are *highlighted*).

---

> > > ### Author Response · Authors · 2025-11-18
> > > **Official Comment by Authors**
> > >
> > > Thank you for your recognition of our work and for raising the score. If you have any questions, we will be more than happy to receive them at any time.

---

### Official Review · Reviewer_S7zc · 2025-10-30

**Soundness:** 4
**Presentation:** 3
**Contribution:** 3
**Rating:** 6
**Confidence:** 4

**Summary:**

This paper introduces a novel framework for interpreting Large Language Models (LLMs) by localizing capability-specific neurons in Feed-Forward Networks, which exhibit strong cohesion and separability. The authors propose the Detecting Capability-Specific Neurons (DCSN) method. validated via enhancement and erasure experiments on models like LLaMA-2-7B. Key contributions include: (1) first demonstration of compositional generalization in these neurons during multi-operator tasks,  (2) Capability Neuron-Level Fine-tuning (CNLF) for efficient, controllable multi-capability boosts, and (3) interpretable insights into LLM invocation mechanisms, advancing beyond overlapping knowledge/task neurons

**Strengths:**

1. Proposes a simple yet effective detection method (DCSN) with interpretable metrics (cohesion & separability).
2. Demonstrates strong empirical evidence for compositional generalization at the neuron level.
3. Introduces efficient neuron-level fine-tuning (CNLF) that improves capability performance with minimal parameters.
4. Clearly defines and isolates capability-specific neurons, moving beyond prior “knowledge/task neuron” work.

**Weaknesses:**

1. All experiments are conducted on relatively small and mid-sized models (up to 13B parameters), so scalability to larger LLMs remains untested.
2. The detection threshold for neuron identification (σ = 6) is arbitrarily chosen, without robustness or sensitivity analysis.
3. Downstream evaluation tasks are few and domain-specific, providing weak external validation beyond arithmetic reasoning.
4. The assumption that capability-specific neurons are fully separable is oversimplified, as inter-capability overlap is observed but not deeply analyzed.
5. The work lacks theoretical grounding to explain why such neurons emerge or how they mechanistically interact to yield compositionally.

**Questions:**

1. How well do the DCSN neuron identification results generalize to more complex or fuzzy capabilities beyond arithmetic, such as reasoning, summarization, or memorization?
2. are there any mechanistic insight into why capability-specific neurons emerge and how they interact across layers to support compositional generalization?
3. Could other attribution methods, such as Integrated Gradients or other feature attribution techniques, be used for detecting capability-specific neurons, and how might their results compare to DCSN?.
4. I suggest adding the following recent works to the Related Works section for better context: [1, 2, 3]. These works focus on neuron-level interventions in LLMs. detecting and pruning neurons that harm generalization or cause copying bias, and are directly relevant to this paper’s theme.



[1] Detecting and Pruning Prominent but Detrimental Neurons in Large Language Models.

[2] Mitigating Copy Bias in In-Context Learning through Neuron Pruning.

[3] Finding Safety Neurons in Large Language Models.

---

> ### Author Response · Authors · 2025-11-17
> **Official Comment by Authors**
>
> Thank you for your constructive feedbacks on the paper! We add detailed explanations for the questions asked in the review.
>
> **Q1:** Experiments on larger models and the selection of thresholds
>
> **W1:** First, due to space constraints, we have provided additional experimental results in the appendix. Specifically, we present the performance of larger models (e.g., Llama2 70B) in Appendix D.3. Additionally, we provide ablation experiments on threshold selection in Appendix D.2.
>
> ---
>
> **Q2:** How well do the DCSN neuron identification results generalize to more complex or fuzzy capabilities beyond arithmetic, such as reasoning, summarization, or memorization?
>
> **W2:** We have conducted experiments on higher-level capabilities, and the results demonstrate the rationality of our capability-specific neurons. First, in Section 6, we performed experiments on three types of capabilities—mathematics, programming, and language. The results show that capability neurons exhibit excellent separability and aggregation, and our fine-tuning method for capability neurons has significant generalization. Furthermore, in Appendix E, we offer additional discussions beyond arithmetic capabilities, fully elaborating on the paper’s logic from arithmetic operators to diverse capabilities, and providing a detailed description of the definition of model capabilities.
>
> ---
>
> **Q3:** Are there any mechanistic insights into why capability-specific neurons emerge and how they interact across layers to support compositional generalization?
>
> **W3:** This is an extremely in-depth question. Based on experimental results, we are the first to discover the compositional generalization phenomenon among capability neurons, which holds significant implications for the field of interpretability. However, to the best of our knowledge, there is currently no clear mechanism in the interpretability field explaining cross-layer interaction issues. We will consider this possibility in future research.
>
> ---
>
> **Q4:** Could other attribution methods, such as Integrated Gradients or other feature attribution techniques, be used for detecting capability-specific neurons, and how might their results compare to DCSN?
>
> **W4:** Thank you for your keen observation. Our localization method is chosen for its effectiveness. First, methods used in previous work [4] and knowledge editing have shown unsatisfactory performance in separability and aggregation experiments, which has been proven in [5]. The neurons localized using DCSN exhibit superior analytical performance and aggregation, leading us to consider this a reasonable research approach. Additionally, we plan to conduct supplementary experiments in future work to compare the existing localization method with those used in [4] and knowledge editing, thereby improving the completeness of the paper.
>
> ---
>
> **Q5:** I suggest adding the following recent works to the Related Works section for better context: [1, 2, 3].
>
> **W5:** Thank you for your suggestion. We commit to adding citations of these works in the revised version, which will enhance the completeness of the paper.
>
>
> **References**
>
> [1] Detecting and Pruning Prominent but Detrimental Neurons in Large Language Models.
>
> [2] Mitigating Copy Bias in In-Context Learning through Neuron Pruning.
>
> [3] Finding Safety Neurons in Large Language Models.
>
> [4] Memory-Based Model Editing at Scale.
>
> [5] Capability Localization: Capabilities Can be Localized rather than Individual Knowledge.

---

> > ### Comment · Reviewer_S7zc · 2025-11-18
> > **Thank you for providing clarifications for my raised issues**
> >
> > Thank you for providing clarifications for my raised issues.
> >
> > I will keep my score unchanged.

---

> > > ### Author Response · Authors · 2025-11-19
> > > **Thank you for your appreciation and recognition of our work**
> > >
> > > Thank you for your appreciation and recognition of our work. We promise to update the above content in the revised version. If you have any questions, please feel free to ask again.

---

### Official Review · Reviewer_1zrH · 2025-11-01

**Soundness:** 2
**Presentation:** 2
**Contribution:** 2
**Rating:** 4
**Confidence:** 4

**Summary:**

This paper aims to understand the effects of neuron composition within large language models (LLM) by identifying neurons linked to specific skills, exploring their compositional generalization, and demonstrating the utility of sparse fine-tuning. The authors first proposed a method called DCSN to detect capability-specific neurons. They curated datasets targeting distinct task capabilities and employed statistical analyses to isolate neurons whose activations are significantly higher for those capabilities. For example, separate neuron groups were found to correspond to addition and multiplication in mathematical reasoning. They further validated the compositional effects of these capacity-specific neurons, confirming that the identified units indeed encode skill-specific functions (Cohesiveness, Separability metrics). Finally, to showcase the practical implications, the authors applied their method to locate neurons associated with mathematics, programming, and other language tasks. Leveraging these neurons for sparse fine-tuning (i.e., CNLF method in this paper) led to superior downstream performance compared to both full-parameter and LoRA fine-tuning, despite involving far fewer trainable parameters.

**Strengths:**

1. The concept of compositional generalization of capability neurons is inspiring. Although previous studies have explored knowledge neurons [1] and capability neurons [2], the combinatorial effects between them, and their implications for model fine-tuning have rarely been investigated. This work provides valuable insights into understanding the inner workings of LLMs at the neuron level (e.g., the mutual interactions between neurons responsible for addition and subtraction).
2. The proposed metrics of cohesiveness and separability offer effective tools for assessing the accuracy and reliability of neuron localization.
3. The demonstrated effectiveness of the CNLF fine-tuning method is impressive, as it enables model training under resource-constrained conditions. The experiments show that fine-tuning only a small subset of neurons can achieve comparable or even superior performance to full-model fine-tuning on tasks such as mathematics.
4. The paper is well written and structured, forming a coherent narrative that progresses from neuron localization to understanding compositional effects, and finally to neuron-level fine-tuning.

**Weaknesses:**

1. Most of the experiments related to mathematical capabilities focus on well-defined basic arithmetic operations. However, the concept of capability neurons may extend far beyond **elementary arithmetic**. Even within mathematics, could “mathematical capability” be decomposed into more **complex sub-capabilities**, such as logical reasoning or symbolic manipulation? It may be worthwhile to further analyze neurons corresponding to these higher-level skills.
2. Related to the first point, the evaluation datasets could be more challenging. GSM8K is relatively basic. Would the localization and fine-tuning of neurons responsible for arithmetic operations (`+, -, *, /`) yield consistent results and efficiency when applied to more difficult benchmarks such as AIME24 or AIME25?
3. The definition of Eq. 3 requires refinement. It currently describes the intersection of sets rather than a quantifiable metric (e.g., a ratio or percentage). However, the subsequent discussion of cohesiveness and separability suggests that these should indeed be ratio-based measures.
4. The main text introduces a hyperparameter $\alpha$, while Appendix D.2 presents an ablation study involving a hyperparameter $\beta$. However, $\beta$ is never defined or mentioned in the main text. It seems likely that $\beta$ corresponds to $\alpha$ in Eq. 2, but this should be clarified explicitly.

**Questions:**

1. How these findings might generalize to other architectures, particularly Mixture-of-Experts [3] models. For example, would specific capabilities tend to cluster within certain experts?
2. This study primarily focuses on neurons within the FFNs. Have you considered the role of attention heads [4] in realizing these capabilities, especially in combining different ones? Is it possible that certain capabilities emerge from the synergistic interaction between FFNs and the attention mechanism?

**Reference**

[1] Dai, D., Dong, L., Hao, Y., Sui, Z., Chang, B., & Wei, F. (2022). Knowledge Neurons in Pretrained Transformers. In Proceedings of the 60th Annual Meeting of the Association for Computational Linguistics (Volume 1: Long Papers) (pp. 8493-8502).

[2] Song, R., He, S., Jiang, S., Xian, Y., Gao, S., Liu, K., & Yu, Z. (2024). Does Large Language Model Contain Task-Specific Neurons?. In Proceedings of the 2024 Conference on Empirical Methods in Natural Language Processing (pp. 7101-7113).

[3] Cai, W., Jiang, J., Wang, F., Tang, J., Kim, S., & Huang, J. (2025). A survey on mixture of experts in large language models. IEEE Transactions on Knowledge and Data Engineering.

[4] Yin, K., & Steinhardt, J. Which Attention Heads Matter for In-Context Learning?. In Forty-second International Conference on Machine Learning.

---

> ### Author Response · Authors · 2025-11-17
> **Official Comment by Authors**
>
> Thank you for recognizing the importance, effort in method, and applications of our work. We outline our response to the main concerns:
>
> **Q1:** Could "mathematical capability" be decomposed into more complex sub-capabilities, such as logical reasoning or symbolic manipulation?
>
> **W1:** Thank you for your constructive suggestion. First, this suggestion was indeed one of our previous considerations. Since it is difficult to fully enumerate mathematical capabilities, our experiments first focus on simple and specific capabilities, and then extend to mathematical capabilities as a whole. Specifically, the paper visualizes the model's mathematical capabilities through simple operators ( "+", "-", "*" and "/") to help readers better understand. Based on the interesting findings from the operator experiments, we extended the entire research to non-operator tasks. In addition, the reviewer's suggestion is meaningful. We are currently organizing data on logical reasoning and symbolic manipulation, and the specific experimental results will be updated in the revised version.
>
> ---
>
> **Q2:** The evaluation dataset could be more challenging, such as AIME24 or AIME25.
>
>
> **W2:** We have conducted relevant experiments, and the results are as follows:
>
> |Qwen3-8B|AIME24|AIME25| Avg. |
> |----  |----  |----  |----  |
> |Original|0.21|0.24|0.23|
> |Ours|**0.27**| **0.28**| **0.28**|
>
> The experimental results show that our method still demonstrates significant effectiveness on datasets such as AIME24 and AIME25, which is consistent with the claims and expectations of the paper.
>
> ---
>
> **Q3:** Regarding the improvement of Formula 3.
>
> **W3:** Thank you for your reminder. We have improved Formula 3 in the revised version and presented it in the form of a ratio.
>
> ---
>
> **Q4:** Regarding the hyperparameter $\beta$.
>
> **W4:** This is a typo, which corresponds to $\sigma $ of  Line 208 in the paper. Thank you for your reminder.
>
> ---
>
> **Q5:** How can these findings be generalized to other architectures, especially Mixture of Experts (MoE) models?
>
> **W5:** This is an interesting suggestion. First, our localization method is also applicable to MoE models and is architecture-agnostic. In addition, we conducted experiments on mathematical capabilities using the Mistral 8 X 7B  MoE model:
>
> |Layers|1~5|6~10| 11~15 | 16~20 | 21 ~25| 26~30| 31~32|
> |----  |----  |----  |----  |----  |----  |----  |----  |
> |Neuron Ratio| 0.17| 0.18| 0.14| 0.14| 0.14|0.15|0.08|
> |Standard Deviation| 156| 182| 145| 163| 169|173|164|
>
> The ‘Standard Deviation’ represents the standard deviation of the call frequency of each expert in each layer.
> The results show that these neurons are scattered and not limited to a few experts. Thank you again for the constructive suggestion; this finding is meaningful.
>
> ---
>
> **Q6:** Regarding the consideration of Self-Attention for localization.
>
> **W6:** Frankly speaking, this was one of our previous considerations, but the experimental results show that it is unreasonable. First, previous studies [1][2] have shown that Self-Attention layers play a connecting role, while FFN layers serve a storage function. In addition, our experiments and [3] indicate that the model's capabilities have the potential for parameter localization in FFN layers rather than in Self-Attention layers. We plan to add a discussion on this possibility in the discussion section.
>
> **References**
>
> [1] Locating and editing factual associations in gpt.
>
> [2] Towards understanding multi-task learning (generalization) of llms via detecting and exploring task-specific neurons.
>
> [3] Capability Localization: Capabilities Can be Localized rather than Individual Knowledge.

---

### Official Review · Reviewer_dWKT · 2025-11-09

**Soundness:** 3
**Presentation:** 2
**Contribution:** 3
**Rating:** 4
**Confidence:** 4

**Summary:**

The paper presents an attempt to identify “capability” specific neurons through correlation of individual neuronal activations and the model’s output tokens. The authors identify about 5% of the top contributing neurons in the FFN layer of the transformer layers for each arithmetic operation. The paper conducts several evaluations to establish the benefits of identifying capability-specific neurons including compositional behavior, erasure, and fine-tuning methods.

Overall, while the paper provides some compelling results, the draft leaves out various questions related to choices made in the work as well as the broader implications of the work.

**Strengths:**

Approach to ranking neurons is simple, although the authors do not provide any sensitivity to the number of samples needed to identify the neurons.

Performance on enhancement, separability and cohesion evaluations show validity of the some of claims made in the work.

Extension to non-mathematical tasks are interesting, although this evaluation is quite limited.

**Weaknesses:**

I. The authors primarily considered FFN layers of the transformer layers. While this may have allowed operation simplicity, it is unclear why the self-attention neurons are ignored in the current analysis. The authors do not provide a strong justification for this choice.

II.  The large part of the work dwells significantly on the arithmetic operations of “+”, “-”, “*” and “/”. It is only in the very last part of the results that the authors provide experimental evidence for other non-arithmetic tasks. Several questions pertain to these experiments. Are the neurons identified similar to the arithmetic operations.

III. Tasks like emotion recognition and coding involve multiple atomic tasks like, semantic parsing, logical understanding, contextual modeling and function reasoning. It is not clear how these are modeled in Section 6.4 (Table 6). The discussion on this part is rather slight, leaving much of the details to the imagination of the reader.

IV. Various modes of introducing arithmetic operations are discussed in Table 1. It is not clear how these modes of introduction change the neuronal activity pattern, for example, if DG is used for identifying the neuronal patterns, does it also generalize to other types of invoking arithmetic operations.

V. Several choices are made throughout the work, without a strong justification. For example, choices like capability specific neurons of 0.05% are used, and in the experiments with erasure 10% of capability neurons are disabled. Secondly, in some of the Tables which report experiments, different models are used for enhancement and erasure experiments (like Table 2). Why do we not have the same model used for both enhancements and erasures. These choices, without a clear articulation, raises significant questions about the generalization of the behaviors reported in the work.

VI. The paper suffers from a lack of novelty. Identifying neural activity using correlation patterns is the only choice considered, while prior works (like Leng et al [2025]) have advocated the use of gradient based relevance scores, while knowledge editing efforts use different similarity metrics and measures.

VII. Further, the authors have focused on individual neurons versus layers or chains of neurons. The authors do not provide any experimental comparison with other prior works which have proposed layers of connected neurons or those with other measures. This makes the experimental comparisons very shallow and do not provide sound justification for the significance of this work.

**Questions:**

As noted in the weakness, several aspects can be elaborated and explored to strengthen the work.
Providing more comparative experimental evidences with other works that have proposed identifying task specific neurons and layers.

Justifying many of the choices made the work, including the motivation for the correlation based metrics used.

Discussing various experiments in detail to highlight the aspects to related applications in non-arithmetic tasks.

Measuring sensitivity and robustness of the neuron identification task when the number of samples are limited.

Improving the limitations sections - it is currently in Appendix with the only mention of computational choices made in terms of the LLMs considered in the work.

---

> ### Author Response · Authors · 2025-11-17
> **Official Comment by Authors[1]**
>
> Thank you for your valuable feedback and for recognizing the novelty of the our method. Below, we address some of the weaknesses raised:
>
> **Q1:** The authors primarily considered FFN layers of the transformer layers.
>
> **W1:** This is the experimental setup based on previous studies. First, prior research [1][2] has shown that Self-Attention layers play a connecting role, while FFN layers serve a storage function. Additionally, [3] indicates that the model's capability has the potential for parameter localization in FFN layers.
>
> ---
>
> **Q2:** The large part of the work dwells significantly on the arithmetic operations of “+”, “-”, “*” and “/”.
>
> **W2:** This is determined by the description method adopted in the paper. The paper visualizes the model's capabilities through simple operators, helping readers better understand the content. Based on interesting experimental findings regarding operators, we extended the entire research to non-operator tasks. Due to space limitations, the description of non-operator tasks in the main text is relatively concise, but we have provided detailed elaboration and discussion in Appendices D and E. Furthermore, we commit to offering more detailed explanations for the concise descriptions and correcting this point in the revised version.
>
> ---
>
> **Q3:** Tasks like emotion recognition and coding involve multiple atomic tasks like, semantic parsing, logical understanding, contextual modeling and function reasoning.
>
> **W3:** This is an interesting insight. First, it is worth noting that the purpose of introducing different task types is to enhance the generalization and persuasiveness of the experiments. In addition, following the reviewers' suggestions, we conducted relevant experiments to explore indicators such as the separability and aggregation of neurons identified using different task types.
>
> |LLaMa-2-7B (+)|DG & TF|DG & MQ|TF & MQ| Avg. |
> |----  | ----  | ----  | ----  | ----  |
> |Cohesiveness| 96.3 | 94.5 |97.2 | 96.0|
> |Separability| 6.3|5.4 | 8.6|6.7 |
>
> The results show that neurons identified through different task types exhibit excellent separability and aggregation, which is consistent with the expectations for capability neurons in the paper. We sincerely thank the reviewers for this suggestion, as it helps improve the completeness of the paper.
>
> ---
>
> **Q4:** Several choices are made throughout the work, without a strong justification.
>
> **W4:** These choices are based on previous research. First, indicators such as 0.05% and 10% in the paper are adopted from the experiments in [3]. To ensure fair comparison with previous work, we retained these indicators. Additionally, regarding the selection of different models, we tested multiple models during the experiment, and their conclusions are consistent with the compositional generalization phenomenon mentioned in the paper. To demonstrate the generalization of this finding, we present results from four models in Table 2. We apologize for any misunderstanding this may have caused to the reviewers and will update this part in the revised version to provide specific experimental results for each model.
>
> ---
>
> **Q5:** Identifying neural activity using correlation patterns is the only choice considered.
>
> **W5:** Thank you for your keen observation. Our localization method is chosen for its effectiveness. First, methods used in previous work [4] and knowledge editing have shown unsatisfactory performance in separability and aggregation experiments, which has been proven in [3]. The neurons localized using correlation patterns in our study exhibit superior analytical performance and aggregation, leading us to consider this a reasonable research approach. Furthermore, we plan to conduct supplementary experiments in future work to compare the existing localization method with those used in [4] and knowledge editing, thereby improving the completeness of the paper.
>
> ---
>
> **Q6:** The authors have focused on individual neurons versus layers or chains of neurons.
>
> **W6:** First, paper [3] has shown that the localization hypotheses for parameter chains and parameter layers are unreasonable, as they exhibit poor separability and aggregation. The motivation for our localization of capability neurons stems from the limitations of previous knowledge localization methods, which is also one of the contributions of our paper. However, the reviewers' suggestion is meaningful—our current experiments have not considered layers or chains of capability neurons, and we will conduct supplementary experiments to explore this possibility.

---

> ### Author Response · Authors · 2025-11-21
> **Official Comment by Authors[2]**
>
> **Q7:** The innovation of the paper.
>
> **W7:** First, our DCSN is a localization method based on capability-specific neurons, which has not been proposed in previous works. Additionally, the core contributions and innovations of this paper lie in revealing the capability invocation mechanism inside the model and discovering the combinatorial generalization phenomenon of capability-specific neurons, which are of great significance for the field of interpretability. Furthermore, we elaborate on the innovations of the paper in detail:
> - **Indicator Innovation**: We are the first to adopt aggregation and separation as accuracy indicators for neuron localization and apply them to cross-comparison of different localization methods. This innovation reveals that previous methods [2,5] targeting knowledge or tasks for localization are seriously flawed, and proves the rationality of using abilities as localization objects. Methods that take knowledge or tasks as localization objects cannot predict our experimental conclusions and have not discovered the phenomenon of combinatorial generalization.
> - **Phenomenon Discovery**: We first discovered the phenomenon of combinatorial generalization in capability-specific neurons. By using four operators (+, -, *, /), we determined the capability call mechanism within the model. When different capability-specific neurons are activated in combination, complex tasks (such as 3-OP) can be accurately solved. When capability-specific neurons are enhanced, the model can generalize well across datasets and tasks (Table 6). This explains the confusion in previous work [2]: enhancing neurons for specific knowledge or tasks leads to a decline in the performance of other knowledge or tasks and poor generalization, because their preliminary assumption (that knowledge or tasks can be parameter-localized) is wrong.
> - **Method Universality**: We discovered the phenomenon of combinatorial generalization using four operators, and this conclusion is not limited to mathematical capabilities. We also identified capability-specific neurons for language and programming in Section 3.4 and Experiment 6. However, existing datasets are insufficient in distinguishing capabilities, while our method can effectively address this challenge. By leveraging the phenomenon of combinatorial generalization of capability neurons for fine-tuning, we can not only significantly improve model performance but also achieve good generalization (Table 6). The discovery of the phenomenon of combinatorial generalization will broadly assist future neuron fine-tuning tasks. This finding is highly novel and practical, which has not been proposed in previous work, and our experimental conclusions cannot be accurately predicted.
>
> **References**
>
> [1] Locating and editing factual associations in gpt.
>
> [2] Towards understanding multi-task learning (generalization) of llms via detecting and exploring task-specific neurons.
>
> [3] Capability Localization: Capabilities Can be Localized rather than Individual Knowledge.
>
> [4] Memory-Based Model Editing at Scale.
>
> [5] Editing models with task arithmetic.

---

### Author Response · Authors · 2025-11-25

The discussion period has lasted for approximately two weeks. We would like to thank Reviewer 3GuE for raising their score to 6, and Reviewer S7zc for confirming their satisfaction with our response. We sincerely invite Reviewers dWKT and 1zrH to inform us whether their concerns have been addressed. Should further clarification be needed, we are happy to provide additional details. Thank you again for your time, thoughtful feedback, and engagement with our work!

---

### Meta-Review · Area_Chair_ArxA · 2025-12-29

**Summary:**

The paper proposes a method (DCSN) to detect "capability-specific neurons" in LLMs and introduces a fine-tuning approach (CNLF) based on these findings. The authors utilize a synthetic dataset of arithmetic operations to argue that these neurons exhibit "compositional generalization" (e.g., activating addition and multiplication neurons simultaneously). While the rebuttal was responsive, like adding experiments on larger models (Llama-70B), different architectures (MoE), and harder tasks, the core mechanistic insight remains heavily tethered to simple arithmetic. The generalization of this "compositional" mechanism to complex reasoning tasks is not rigorously established, and the methodological choices (e.g., focusing solely on FFNs) lack sufficient theoretical justification.

**Reviewer Concerns:**

Addressed by Rebuttal:

- Scalability and Architecture: Reviewers (S7zc, 1zrH) questioned if the findings hold for larger models or MoEs. The authors successfully addressed this by providing additional results on Llama-2-70B and Mistral 8x7B (MoE) in the appendix.

- Narrow Experimental Domain: Reviewer 3GuE initially criticized the paper for restricting analysis to arithmetic operators. The authors clarified that later sections cover Math and Coding tasks, leading this reviewer to raise their score.


Still Outstanding:

- Depth of Mechanistic Insight: Despite the additional experiments, the paper's central scientific claim that compositional generalizationis only rigorously verified in the "toy" domain of arithmetic operators. The extension to complex domains (coding/reasoning) demonstrates performance improvements but fails to prove that the same compositional mechanism is at play, leaving the core hypothesis unproven for general LLM capabilities (Reviewer dWKT).

- Theoretical Grounding: Reviewer S7zc noted the lack of a theoretical framework explaining why these neurons emerge or how they interact across layers. This remains an open empirical observation without a solid mechanistic explanation.

- Methodological Justification: Reviewer dWKT raised valid concerns regarding the exclusive focus on FFNs (ignoring Self-Attention) and the specific selection of hyperparameters (e.g., detection thresholds). The authors' response relied heavily on citing prior work rather than providing rigorous ablations or theoretical justifications within the context of this specific study.

**Reviewer Scores:**

- Reviewer dWKT (4): Likely remains unconvinced. The reviewer questioned the fundamental choices (FFN only, specific thresholds) and the depth of the non-arithmetic evaluation. The authors' response justified these choices based on prior work rather than new rigorous ablation, which may not be sufficient to change the score.

- Reviewer 1zrH (4): Although the authors added experiments on MoE and AIME, the fundamental concern about whether "mathematical capability" can be simply decomposed and localized as proposed likely remains. The reviewer might view the "compositional" claim as overclaimed for complex reasoning tasks.

- Reviewer 3GuE (2 -> 6): This reviewer raised their score after noticing the additional experiments. However, their initial critique regarding the "narrow domain" of the core analysis remains a valid point for the AC to consider regarding the paper's significance.

- Reviewer S7zc (6): Supported the paper but noted the lack of theoretical grounding for why these neurons emerge, which remains an open issue.

---

### Decision · Program_Chairs · 2026-01-26

Reject